# A Systems Biology Approach for Addressing Cisplatin Resistance in Non-Small Cell Lung Cancer

**DOI:** 10.3390/jcm12020599

**Published:** 2023-01-11

**Authors:** Sravani Ramisetty, Prakash Kulkarni, Supriyo Bhattacharya, Arin Nam, Sharad S. Singhal, Linlin Guo, Tamara Mirzapoiazova, Bolot Mambetsariev, Sandeep Mittan, Jyoti Malhotra, Evan Pisick, Shanmuga Subbiah, Swapnil Rajurkar, Erminia Massarelli, Ravi Salgia, Atish Mohanty

**Affiliations:** 1Department of Medical Oncology and Therapeutics Research, City of Hope National Medical Center, Duarte, CA 91010, USA; 2Department of Systems Biology, City of Hope National Medical Center, Duarte, CA 91010, USA; 3Translational Bioinformatics, Center for Informatics, Department of Computational and Quantitative Medicine, City of Hope National Medical Center, 1500 Duarte Rd, Duarte, CA 91010, USA; 4Department of Pathology, University of California, La Jolla, San Diego, CA 92093, USA; 5Montefiore Medical Center, The University Hospital for Albert Einstein College of Medicine, Bronx, NY 10467, USA; 6Department of Medical Oncology and Therapeutics Research, City of Hope National Medical Center, 1000 FivePoint, Irvine, CA 92618, USA; 7Cancer Treatment Centers of America (CTCA) Chicago, 2520 Elisha Avenue, Zion, IL 60099, USA; 8Department of Medical Oncology and Therapeutics Research, City of Hope National Medical Center, 1250 S. Sunset Ave., Suite 303, West Covina, CA 91790, USA; 9Department of Medical Oncology and Therapeutics Research, City of Hope National Medical Center, 1100 San Bernardino Road, Suite 1100, Upland, CA 91786, USA

**Keywords:** drug resistance, cisplatin, non-small cell lung cancer, group behavior, IDPs, phenotype switching, mathematical modeling

## Abstract

Translational research in medicine, defined as the transfer of knowledge and discovery from the basic sciences to the clinic, is typically achieved through interactions between members across scientific disciplines to overcome the traditional silos within the community. Thus, translational medicine underscores ‘Team Medicine’, the partnership between basic science researchers and clinicians focused on addressing a specific goal in medicine. Here, we highlight this concept from a City of Hope perspective. Using cisplatin resistance in non-small cell lung cancer (NSCLC) as a paradigm, we describe how basic research scientists, clinical research scientists, and medical oncologists, in true ‘Team Science’ spirit, addressed cisplatin resistance in NSCLC and identified a previously approved compound that is able to alleviate cisplatin resistance in NSCLC. Furthermore, we discuss how a ‘Team Medicine’ approach can help to elucidate the mechanisms of innate and acquired resistance in NSCLC and develop alternative strategies to overcome drug resistance.

## 1. Introduction

Cancer is one of the major contributors to global mortality. According to Cancer Statistics, which is published every year [1], 1.9 million new cancer cases and 609,360 deaths from cancer are estimated in 2022 in the US alone, which is about 1670 deaths a day. Among all the cancer types, lung cancer (both small and non-small cell) is the most prevalent, and it is estimated that a total of 236,740 people will be diagnosed with lung cancer in 2022, which is 1 in 16 people in the US alone [2,3,4,5]. Lung cancer is generally a disease of middle-aged and elderly smokers, usually with several comorbid smoking-related conditions, such as emphysema, chronic bronchitis, widespread atherosclerosis, and degenerative disorders of the central nervous system (CNS) and other organs [6,7,8]. Despite significant developments in preventing, screening, and treating lung cancer over the past decade, innate and acquired resistance to chemotherapeutic agents and radiation remains a vexing problem, and success in increasing the life expectancy of patients is limited. 

A vast majority (~85%) of lung cancer patients have a group of histological subtypes collectively known as non-small cell lung cancer (NSCLC). Among the various subtypes, lung adenocarcinoma (LUAD) and lung squamous cell carcinoma (LSCC) are the most common subtypes [9]. LUAD is mostly driven by driver oncogenes, such as EGFR, KRAS (G^12^C), MET, ALK, etc., against which targeted therapies are available. On the other hand, no known targetable driver oncogenes have been identified for LSCC; thus, the therapeutic options for LSSC patients are limited. 

NSCLC patients are offered a broad range of genotoxic drugs, such as cisplatin or carboplatin alone or in combination with immunotherapy [10], and often respond to treatment initially. However, most patients develop drug resistance, and numerous mechanisms underlying drug resistance have been identified, mostly in preclinical models of the disease [11,12,13]. Unfortunately, many of these mechanisms do not always hold in vivo and, very often, are not effective in the clinic even if they appear promising in the in vivo models. Therefore, a collaborative effort integrating the preclinical studies with clinical outcomes could help to better understand the mechanism of drug resistance. 

Here, we summarize the concept of ‘Integrating Clinical and Translational Research Networks—Building Team Medicine’ from a City of Hope perspective (Figure 1). Using cisplatin resistance in NSCLC as a paradigm, we describe how basic research scientists with expertise in fields as varied as cancer biology, cell and molecular biology, biochemistry, biophysics, structural biology, and mathematical and computational biology; clinical research scientists; and medical oncologists working together with a true ‘Team Medicine’ spirit, uncovered a non-genetic mechanism underlying cisplatin resistance in NSCLC and identified carfilzomib (CFZ), a previously approved proteasome inhibitor, to alleviate resistance. The team was led by the Department Chair, a thoracic oncologist who helped coordinate the team’s efforts, much like a tumor board that comprises clinical specialists, nurses, and care providers does in a hospital setting. Furthermore, we discuss how this ‘Team Medicine’ approach also helped explore novel treatment strategies that could potentially preclude, attenuate, or at least delay, the onset of cisplatin resistance in these patients. 

## 2. Mechanisms of Drug Resistance

Drug resistance is the major obstacle to long-term patient survival [14]. Cancer cells can escape therapy and exhibit drug resistance by different routes and many of these routes remain unpredictable and difficult to characterize [15]. A better understanding of the molecular mechanisms that help in tumor progression and drug resistance is essential in designing cancer subtype specific treatments. Drug resistance is defined as the inheritable ability of the cells to survive clinically relevant drug concentrations [16]. Drug resistance can either pre-exist before the start of the treatment, referred to as innate resistance, or develop in response to the treatment and is referred to as acquired resistance [15,16,17,18,19]. Innate drug resistance is typically thought to involve genetic mutations, while acquired resistance is generally believed to be due to both genetic and non-genetic/epigenetic changes. In either case, resistance to therapy is associated with metastatic disease and poor survival rates in patients [16,20,21,22]. Regardless, however, numerous mechanisms [23,24] that promote drug resistance have been reported in the literature, adding to the resistance conundrum (Figure 2). 

Cancer drug resistance is multi-factorial and not solely driven by genetic mechanisms [25]. In fact, an increasing body of evidence shows that non-genetic mechanisms, such as lineage plasticity [26] (change in cell identity), epigenetic factors that regulate gene expression, and phenotype plasticity, contribute to cancer drug resistance [27]. Cancer cells escape the drug assault by two phenomena; ‘tolerance‘, which is the ability of the cell to survive transient exposure to high drug concentration, and ‘persistence’, which is the ability of a subpopulation of a clonal population to survive exposure to high concentrations of a drug [27]. Drug-tolerant persisters (DTP) remain major factors in cancer relapse and in developing drug resistance [28]. Persistence is observed in low frequency in tumor cells with reduced proliferation rate and metabolism that helps them in tolerating drug insult. The genetic makeup of the DTPs is indistinguishable from the bulk tumor population and the resistance exhibited by them reverts to the sensitive state upon drug removal [29,30,31,32,33,34]. However, Shaffer et al. [35] showed that persisters cells exhibit significant variability at the single-cell level. Furthermore, these variabilities eventually decide the fate of the cell regarding whether to irreversibly become resistant to drug treatment. 

Two phenomena that determine whether a given cancer cell population will undergo non-genetic evolution of drug resistance are epigenetic heterogeneity and epigenetic plasticity [27,36,37,38]. Epigenetic heterogeneity refers to the overall variability in the epigenetic landscape across a given cell population, which is influenced by both cell-intrinsic and cell-extrinsic stimuli. Epigenetic plasticity, on the other hand, is the capacity of a cell to alter its epigenetic state in response to either internal or external stimuli [27,36,37,38]. It is crucial to understand that both epigenetic heterogeneity and epigenetic plasticity are not completely independent variables; for example, various cancer cell types exhibit heterogeneity because the epigenetic state of the population is more plastic.

A stable mechanism of non-genetic resistance can result in the pre-existence of resistant clones in the subpopulation, in which case drug resistance simply emerges through Darwinian selection and is completely dependent on epigenetic heterogeneity. Alternatively, the stable origin of a non-genetic resistance can also be a result of gradual Darwinian or Lamarckian induction (Figure 2).

Recent evidence indicates that the genetic and non-genetic mechanisms of drug resistance are not mutually exclusive but indeed co-exist (meaning that both evolutionary phenomena of Darwinian selection and Lamarckian induction may be active) within a given cancer type and drive the drug resistance that eventually led to therapy failure. The genetic/nongenetic duality as described in the review is believed to be a major contributor to the complexity of drug resistance [39]. Designing drugs that target only the genetic mutations is like playing a whack-a-mole game where the player has zero chance of winning, even after multiple attempts. Thus, it is important to gain a deeper understanding of the relative contributions of genetic and non-genetic mechanisms especially, by understanding how, why, and when these non-genetic alterations occur so that one can hit the desired target consistently. 

In addition to genetic mutations/epigenetic changes, protein interaction networks (PIN) also contribute to drug resistance [40,41,42]. PIN dynamics are orchestrated by the hub proteins, which are typically intrinsically disordered proteins (IDPs). IDPs lack 3D structure but exist as conformational ensembles. Indeed, ~80% of cancer-associated proteins, for example, p53, cyclins, MYC, SOX2, paxillin, etc., are IDPs [43]. This article focuses on acquired resistance to cisplatin resistance in NSCLC.

## 3. Cisplatin Resistance in NSCLC

Cisplatin is one of the platinum-based frontline chemotherapeutic agents used to treat solid tumors in a wide spectrum of cancers, including lung, ovarian, colorectal, head and neck, and testicular [44,45,46]. Cisplatin delivers its attack by entering cancer cells and binding to DNA, thus forming DNA adducts. These adducts block transcription and DNA synthesis, which activates the intracellular signal transduction that helps to eliminate the tumor lesions [43]. Patients usually have a good initial response to cisplatin-based chemotherapy but relapse later, because the development of acquired or innate resistance markedly reduces its clinical effectiveness [46,47,48,49,50]. Various molecular mechanisms, such as altered DNA repair and the cellular accumulation of the drug, as well as the cytoplasmic inactivation of the drug, are a few of many pathways through which patients usually develop resistance to cisplatin. The goal of personalized medicine is to develop better responses to the drug in the clinic. Here in this review, we will comprehensively discuss the non-genetic mechanisms of cisplatin resistance in NSCLC [51,52]. Cellular resistance to cisplatin may conceivably be based upon the overexpression or inactivation of certain oncogenes both in genetic and epigenetic pathways [47,49,53]. One such epigenetic mechanism involves focal adhesion complex (FA) and the components that contribute to cisplatin-resistance in NSCLC.

### 3.1. Focal Adhesion and Cisplatin Resistance

Focal adhesions (FAs) are contact points for a cell that interact with the extracellular matrix (ECM) and regulate diverse cellular processes, such as apoptosis, proliferation, migration, and differentiation. The principal components of FAs are integrin, paxillin, focal adhesion kinase (FAK), SRC (Src Oncogene), talin, tensin, vinculin, and actin [54]. Integrins function as transmembrane receptors for extracellular ligands and transduce biochemical signals into the cell. Integrins, when bound to ligands, are shown to be involved in a variety of signaling pathways, such as the cell cycle; the organization of the intracellular cytoskeleton; and in mediating the translocation of new receptors to the cell membranes with an α and a β subunit [55,56]. In mammals, there are 24α and 9β integrins among which integrin β4 (ITGB4) is believed to be unique due to its >1000 amino acid cytoplasmic domain when compared to other β-forms that typically have cytoplasmic domains of ~59 amino acids [57]. Furthermore, the unique property of ITGB4 is that it heterodimerizes with ITGα6 as well as ITGα7 [52,58]. However, ITGβ4 and its role in cisplatin resistance remained poorly understood until recently.

Another important component of the FA complex is paxillin (PXN). Human PXN is a 68-kDa (591 amino acids) protein [59]. The LUAD upregulation of PXN is associated with tumor progression and metastasis [60,61]. The phosphorylation of PXN leads to the activation of the downstream pathways of MAPF/ERK, resulting in cisplatin resistance [50]. PXN contains an N-terminus proline-rich region that anchors SH3-containing proteins along with five leucine-rich residues (LD domains 1–5) with a consensus sequence of LD*X*LLXXL [62,63]. The LD2-LD4 region includes sequences for the recruitment of signaling and structural molecules, such as FAK, vinculin, and Crk [62,64,65]. This region has also been reported to interact with integrin; more specifically, integrin α4 (ITGA4). Interestingly, PXN is an IDP [66]. The C-terminal region of PXN is believed to be involved in anchoring PXN to the plasma membrane and targeting to FA complex. The C-terminal of the FA complex harbor Cysteine-Histidine-enriched Lin11/Isl1/Mec3 (LIM) domains that form zinc fingers, suggesting that PXN could bind DNA and act as a transcription factor [67]. In addition to LD domains, LIM domains contain the SH3 domain and SH2 domain that forms a docking site for many tyrosine and threonine kinases and recruit additional enzymes into the complex, eventually leading to the activation of canonical signaling through the Ras-mitogen-activated protein kinase (MAPK), phosphoinositide-3-kinase (PI3K)-Akt, and phospholipase C-gamma (PLC-γ) pathways. 

Our recent work showed that NSCLC tumor tissue has the heterogenous expression of PXN/ITGB4, and patients with the increased expression of both these genes have poor overall survival [50]. Furthermore, the cell lines that were identified to be cisplatin-resistant were also observed to have elevated levels of ITGB4/PXN. The knocking down of ITGB4 and PXN attenuated cell growth and enhanced apoptosis in 2D and 3D cultures. Interestingly, the double knockdown affected the expression of several genes, including USP1/VDAC1. Chromatin immunoprecipitation revealed a reduced binding of acetylated H3K27 at the promoter region of USP1 on the knocking down of ITGB4/PXN, highlighting the epigenetic regulation of various genes by these two proteins. Further, the knocking down of USP1 and VDAC1 generated a similar phenotype as the knockdown of ITGB4/PXN (Figure 3). The suppression of VDAC1 resulted in increased mitochondrial respiration and the generation of reactive oxygen species, leading to DNA damage, whereas the suppression of USP1 affected the DNA repair caused by adduct formation induced by cisplatin. Thus, these results highlighted the important role of the FA-associated complex-associated genes in cisplatin resistance and suggested that disrupting the interactions between the key components could potentially alleviate cisplatin resistance.

### 3.2. Mathematical Modeling Suggests Bistability Drives Phenotypic Switching

Our work also highlighted the role of ITGB4 in defining tumor heterogeneity. An immunohistochemistry analysis of patient samples and NSCLC cell lines confirmed the differential expression of ITGB4 [52]. Therefore, ITGB4 was used as a marker to sort low and high ITGB4-expressing NSCLC cells. Interestingly, the low ITGB4-expressing cells, after a few days in culture, were able to recreate the heterogeneous population of ITGB4-expressing cells, but the cells sorted for high ITGB4 failed to recreate the heterogeneous expression of ITGB4. These results were suggestive that low ITGB4-expressing cells have more plasticity to recreate the heterogeneity compared to high ITGB4-expressing cells. Further, RNA seq analysis carried out on the ITGB4 knockdown cells suggested a bistable relation between the microRNA 1-3p and ITGB4. A mathematical model developed based on the expression of these two genes indicated bistability; in a mixture of heterogeneous cells, some cells express high ITGB4/low microRNA 1-3p, low ITGB4/high microRNA 1-3p, or an equal expression of both ITGB4 and miRNA 1-3p (intermediary cells). Intermediary cells can shift in either direction depending on the environmental cues, for example, they could increase ITGB4 expression to tolerate cisplatin toxicity, and in absence of a drug, they could return to normal or low expressing subtypes [52].

### 3.3. Novel Alternatives to Alleviate Cisplatin Resistance

To identify small molecules that could potentially disrupt the interaction between PXN and FAK and, hence, perturb the focal adhesion complex and its downstream signaling, we used an in silico screening approach to screen a library of FDA-approved compounds. The screen identified several compounds that were found to sensitize the platinum-resistant NSCLC cells. Of these, carfilzomib (CFZ) was the most efficacious (IC50 in the low nanomolar range) and was able to induce DNA damage and apoptosis in NSCLC cells [68]. Furthermore, CFZ was also found to significantly inhibit migration, wound-healing, and ITGB4 expression at sublethal doses. Altogether, the data revealed an alternative and more efficient approach to treating lung cancer patients with cisplatin resistance. 

### 3.4. Group Behavior and Phenotypic Switching Enable NSCLC Cells to Evade Chemotherapy

Phenotypic plasticity is critical for cancer cells to adapt themselves and survive [69]. Because of phenotypic plasticity, cancer cells are adept at switching their phenotypes in response to either intrinsic or extrinsic (environmental) cues. Thus, phenotypic plasticity enables cancer cells to undergo epithelial-to-mesenchymal transition (EMT) in order to facilitate distant metastasis; switch from being drug-sensitive to becoming drug-tolerant and, eventually, -resistant; or acquire stem cell-like characteristics. These phenomena help cancer cells to adapt to the fitness landscape and withstand drug treatment. Emerging evidence also indicates that both genetic and non-genetic mechanisms play crucial roles in the adaptability or cooperation between cancer cells to withstand stressful conditions. Hata et al. [30] provided clinical evidence showing that drug-resistant cells can both pre-exist and can evolve from drug-tolerant cells. If so, how does the co-existence of drug-sensitive and drug-tolerant/resistant clones impact their ability to cooperate or compete (group behavior) to evade drug toxicity [70,71]?

To address this question, we employed an approach that embodied both experimental methods and mathematical modeling, again using the cisplatin treatment of NSCLC cell lines [53]. Cisplatin-sensitive H23 and cisplatin-tolerant H2009 NSCLC cells were co-cultured and monitored in real-time in order to discern differences in their behavior. The two cell line cultures were grown as either monotypic (grown by themselves) or as heterotypic cultures (co-culture of tolerant and sensitive cells) in different ratios 1:1, 2:1, 4:1, and 8:1 and their growth rates were monitored in real time using a live cell imaging system. The data revealed that the tolerant cell proliferation was suppressed in the presence of sensitive cells at a 1:1 ratio and the proliferation could be rescued by increasing the fraction of tolerant cells in the co-cultures (Figure 4). The experiment was also repeated for the alternative ratios where the sensitive cells were increased, keeping the tolerant cells constant and the same result was observed, i.e., tolerant cell growth was inhibited by sensitive cells; however, the addition of a drug or increase in the ratio of tolerant cells in the population favored the growth of tolerant cells.

Considering the key observations from these in vitro studies, such as (i) sensitive cells inhibiting tolerant cell proliferation in a co-culture in the absence of cisplatin, (ii) the suppressive effect being stronger upon longer incubation compared to shorter incubation, and (iii) the competition by the sensitive cells being eliminated in the presence of cisplatin, a new mathematical approach called the Phenotypic Switch Model with Stress Response (PSMSR) was developed to fit the observed growth curves, the model conglomerate concepts from chemical reaction kinetics, and the cooperative behavior of drug-tolerant phenotypes in the community. A distinguishing feature of the PSMSR model is that it considers the ability of cancer cells to switch phenotypes. In addition to several testable predictions, the most important takeaway from the modeling exercise is that a small population of the tolerant cells may help the drug-sensitive cells to sustain proliferation. However, high levels of or continuous drug treatment, such stress removal mechanisms, may be insufficient to sustaining sensitive cell viability. Thus, it follows that it is essential to turn off phenotypic switching in such situations and allow the sensitive cells to become extinct and the tolerant cells to proliferate, which is the fundamental basis of ‘adaptive’ therapy or intermittent therapy strategy [72,73].

## 4. Conclusions and Future Perspective

In the cancer world, drug attrition rates are notorious—several drugs are effective in preclinical studies but only a few are approved for clinical use [74]. Furthermore, while most approved cancer drugs do help in improving the life expectancy of the patients, cancer cells often develop resistance against these therapies and relapse as resistant and metastatic diseases. Moreover, underlying mechanisms remain poorly understood. 

The prescribed strategy that a physician follows—administering the maximal dose continuously in the shortest possible time—can lead to counterproductive and potentially adverse outcomes, such as drug resistance through genetic and non-genetic mechanisms, as discussed above. This led to exploring the emergence of alternative therapies and approaches [75,76,77,78], for example, adaptive/intermittent therapies [70,79]. The basic principle of intermittent therapy is to administer a lower therapeutic dose of the drug than the maximally tolerated dose to maintain a stable disease. The major advantage of intermittent therapy is an improved quality of life for the patient due to low drug dosage and, thus, fewer side effects. By keeping the drug doses low and intermittent (with drug holidays in between), the proliferation of resistant subclones can be delayed. Some of the success stories of intermittent therapy have been seen in rectal, pediatric sarcoma, prostate, and breast cancer [80,81,82,83,84,85]. 

Based on our observations and those of others from the literature, we believe that maintaining a stable disease may be more prudent. A good example is a study by Klotz et [78], where they treated 20 patients with advanced prostate cancer with intermittent endocrine therapy (diethylstilbesterol in 19 cases and flutamide in 1 case). These patients were treated until a clinical response was demonstrated, with a mean initial treatment duration of 10 months (range 2–70 months). The treatment was then stopped and re-started when tumors relapsed, with mean interval times of 8 months (range 1–24 months). All relapsed patients responded to the re-administration of the drug. Patients had a better quality of life during the drug holidays of the treatment. Indeed, subsequent studies [86], including a meta-analysis (Marlon et al.) [87], also concluded that intermittent androgen deprivation can be considered as an option for recurrent or metastatic prostate cancer.

Therefore, the data from our studies on cisplatin resistance in NSCLC not only lend further credence to the paradigm of intermittent strategy to maintain stable disease but also underscore the nuances and benefits of a ‘Team Medicine’ approach from a systems biology perspective. 

## Figures and Tables

**Figure 1 jcm-12-00599-f001:**
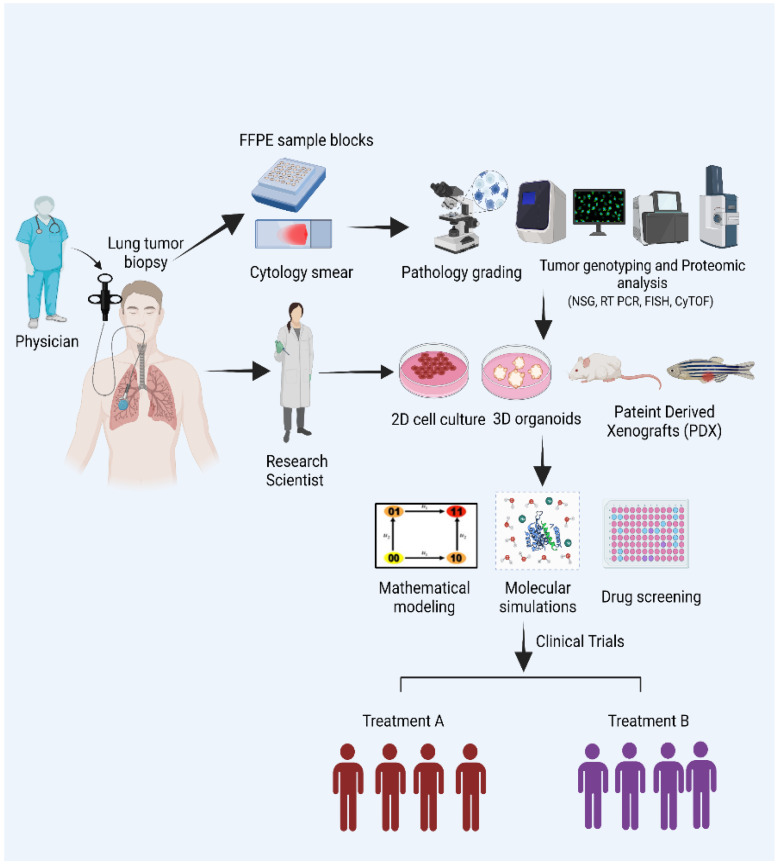
Schematic representing the ‘Team Medicine’ approach.

**Figure 2 jcm-12-00599-f002:**
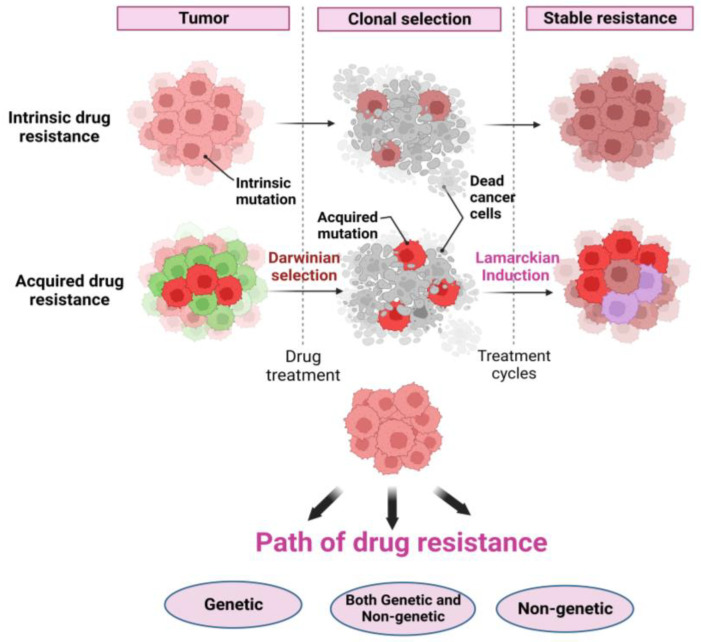
Schematic representation of acquired resistance. Acquired resistance can arise through the Darwinian section or Lamarckian induction. Path of drug resistance where a tumor cell can become resistant purely due to genetic changes through non-genetic alteration in a particular genotype or through initial non-genetic changes combined with genetic mutations.

**Figure 3 jcm-12-00599-f003:**
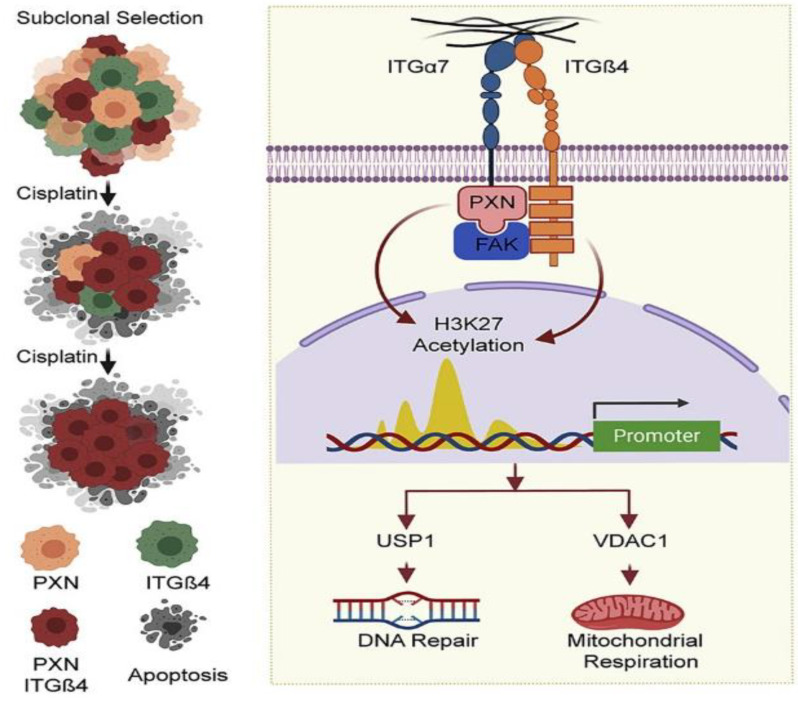
Schematic depicting the interaction between ITGB4 and PXN regulating downstream proteins USP1 and VDAC1 at the transcriptional level to coordinate cisplatin resistance; taken from Reprinted/adapted with permission from Ref. [51].

**Figure 4 jcm-12-00599-f004:**
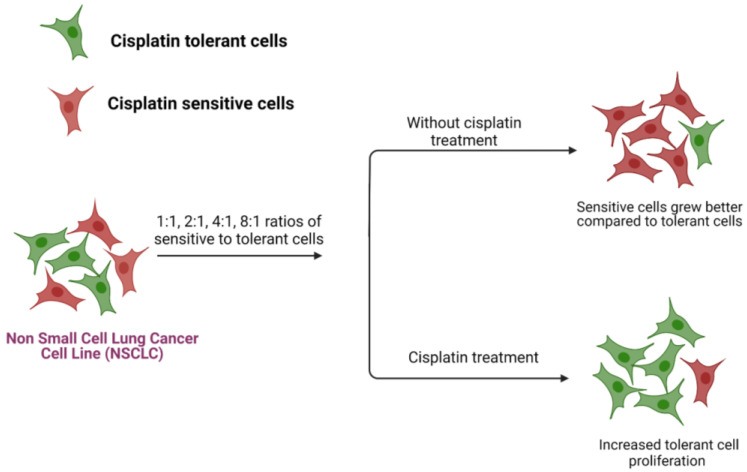
Behavior of cisplatin-sensitive and tolerant NSCLC cells in the 2D co-culture.

## Data Availability

The data are available upon request from the corresponding author.

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
