# Peer review of "A Systems Biology Approach for Addressing Cisplatin Resistance in Non-Small Cell Lung Cancer"

_jcm, 2023, doi:10.3390/jcm12020599_

Round 1

Reviewer 1 Report

Congratulations to the authors for their diligent and brilliant work.

The following are my comments.

First, as you mentioned the critical role and importance of "team medicine", who play the key role (coordinator) in such a team should be clearly defined.

Second, based on your thoughts, rather than cure, stable disease seems to be the favorable status upon treatment efficacy. The exploration of resistance of cancer treatment is truly an unmet need. Nowadays, the trend of anti-cancer treatment is going to the way of precision medicine/individualized medicine. Could you justify your perspective by comparing the relevant literature in an objective manner?

Author Response

  1. First, as you mentioned the critical role and importance of "team medicine", and who plays the key role (coordinator) in such a team should be clearly defined.

Author’s Response: We appreciate the reviewer for bringing up this important point. We have now highlighted this aspect in the manuscript.

  1. Second, based on your thoughts, rather than cure, the stable disease seems to be favorable to treatment efficacy. The exploration of resistance to cancer treatment is truly an unmet need. Nowadays, the trend of anti-cancer treatment is going in the way of precision medicine/individualized medicine. Could you justify your perspective by comparing the relevant literature in an objective manner?

Author’s Response:  This is an excellent suggestion. Accordingly, we have summarized studies from the literature on intermittent therapy for another cancer namely prostate cancer in the revised manuscript.

Reviewer 2 Report

1. In the manuscript, the author mainly describes the research ideas of the research team on acquired cisplatin resistance in non-small cell lung cancer. We can see the innovative thinking of the authors' team in the study of tumor resistance. The model of systematic research combined with a variety of basic medical and engineering methods is worthy of affirmation.

2. As a "team medicine", the ultimate goal of research should be to implement clinical diagnosis and treatment. From the manuscript, we do not have a good understanding of how the author team implemented the clinical work. Can the results of this research approach lead to rapid clinical guidance?

3. The phenomenon of bistability drives phenotypic switching does not necessarily hold true for most cancers. Are there limitations to screening for new treatments for tumors that don't have this phenomenon?

Author Response

Reviewer 2: Comments and Suggestions for Authors

  1. In the manuscript, the author mainly describes the research ideas of the research team on acquired cisplatin resistance in non-small cell lung cancer. We can see the innovative thinking of the authors' team in the study of tumor resistance. The systematic research model combined with a variety of basic medical and engineering methods is worthy of affirmation.

Author’s Response:  We wish to thank the reviewer for the appreciation and encouragement and agree that this is the future of medicine.

  1. As a "team medicine", the ultimate goal of research should be to implement clinical diagnosis and treatment. From the manuscript, we do not have a good understanding of how the author's team implemented the clinical work. Can the results of this research approach lead to rapid clinical guidance?

 Author’s Response: We appreciate the reviewer’s comment. Indeed, the recent development of zebrafish patient-derived xenografts (zPDX) models where the cancer cells are implanted in zebrafish provides rapid, accurate, clinical outcomes to guide personalized treatment for cancer patients with aggressive tumors. zPDX models provide clinicians individualized data on drug treatment and treatment regimens (intermittent or continuous therapy).

3. The phenomenon of bistability drives phenotypic switching does not necessarily hold true for most cancers. Are there limitations to screening for new treatments for tumors that don't have this phenomenon?

Author’s Response:  Bistability is a mathematical term used to describe the steady state of a dynamical system. The phenotype of any cell including a cancer cell can be determined by bistability (or multistability or oscillatory dynamics) either stochastically or in response to a specific environmental cue. Below are a few examples that dwell on this topic; as can be seen, it is not cancer specific. Therefore, there are no limitations to screening for new treatments for tumors. In lung cancer cells, a mathematical model that we developed based on the expression of microRNA 1-3p and ITGB4 indicated bistability that is, in a mixture of heterogeneous cells, some cells will express high ITGB4/low microRNA 1-3p, low ITGB4/high microRNA 1-3p, or equal expression of both ITGB4 and miRNA 1-3p (intermediary cells), and hence exhibit a cisplatin sensitive or tolerant phenotype. In other cases, investigators have also observed bistability based on the expression of genes and the feedback loops specific to those cancers.

  1. Li Y, Li Y, Zhang H, Chen Y. MicroRNA-mediated positive feedback loop and optimized bistable switch in a cancer network Involving miR-17-92. PLoS One. 2011;6(10):e26302. DOI: 10.1371/journal.pone.0026302. Epub 2011 Oct 14. PMID: 22022595; PMCID: PMC3194799.
  2. Singh D, Bocci F, Kulkarni P, Jolly MK. Coupled Feedback Loops Involving PAGE4, EMT, and Notch Signaling Can Give Rise to Non-genetic Heterogeneity in Prostate Cancer Cells. Entropy (Basel). 2021 Feb 26;23(3):288. DOI: 10.3390/e23030288. PMID: 33652914; PMCID: PMC7996788.
  3. Shiraishi T, Matsuyama S, Kitano H. Large-scale analysis of network bistability for human cancers. PLoS Comput Biol. 2010 Jul 8;6(7):e1000851. DOI: 10.1371/journal.pcbi.1000851. PMID: 20628618; PMCID: PMC2900289.